# Investigating the roots of the nonlinear Luttinger liquid phenomenology

L. Markhof[*], M. Pletyukhov, V. Meden

Institut für Theorie der Statistischen Physik, RWTH Aachen University and
JARA–Fundamentals of Future Information Technology, 52056 Aachen, Germany
* lisa.markhof@rwth-aachen.de

October 10, 2019

## Abstract

The nonlinear Luttinger liquid phenomenology of one-dimensional correlated Fermi systems is an attempt to describe the effect of the band curvature beyond the Tomonaga-Luttinger liquid paradigm. It relies on the observation that the dynamical structure factor of the interacting electron gas shows a logarithmic threshold singularity when evaluated to first order perturbation theory in the two-particle interaction. This term was interpreted as the linear one in an expansion which was conjectured to resum to a power law. A field theory, the mobile impurity model, which is constructed such that it provides the power law in the structure factor, was suggested to be the proper effective model and used to compute the single-particle spectral function. This forms the basis of the nonlinear Luttinger liquid phenomenology. Surprisingly, the second order perturbative contribution to the structure factor was so far not studied. We first close this gap and show that it is consistent with the conjectured power law. Secondly, we critically assess the steps leading to the mobile impurity Hamiltonian. We show that the model does not allow to include the effect of the momentum dependence of the (bulk) two-particle potential. This dependence was recently shown to spoil power laws in the single-particle spectral function which previously were believed to be part of the Tomonaga-Luttinger liquid universality. Although our second order results for the structure factor are consistent with power-law scaling, this raises doubts that the conjectured nonlinear Luttinger liquid phenomenology can be considered as universal. We conclude that more work is required to clarify this.

# 1  Introduction

The low-energy properties of interacting fermions confined to one spatial dimension (1d) cannot be described within the Fermi liquid theory. Even if the two-particle interaction does not drive the system out of its metallic phase, e.g., into a Mott insulating one, the fundamental excitations are of collective nature instead of being fermionic quasi-particles. The (continuum) Tomomaga-Luttinger model (TLM) is the metallic low-energy fixed point model of a large class of microscopic models under a renormalization group (RG) flow [1,2]. It plays the same role within universal Tomonaga-Luttinger liquid (TLL) theory [3] as the free Fermi gas does in Fermi liquid theory. Once the dependence of the parameters of the TLM on the ones of a given microscopic model are known, thermodynamic properties and correlation functions at low energy scales can be computed within the TLM. In crucial difference to the free Fermi gas the TLM Hamiltonian contains a two-particle interaction. Still, employing the method of bosonization [1–4] allows to exactly compute essentially all observables and correlation functions of the TLM of interest.

In the construction of the TLM it is assumed that the fermionic single-particle dispersion relation has two strictly linear branches. Any curvature is indeed RG irrelevant and does not affect the low-energy properties [1, 2]. Furthermore, only two types of scattering processes with momentum transfer $|q| \ll k_F$, so-called $g_2$- and $g_4$-processes, are considered [5]. Here, $k_F$ denotes the Fermi momentum. Both steps are crucial to obtain an exact solution employing the constructive bosonization approach [1, 3, 4].

Microscopic models such as the interacting electron gas or the tight-binding chain have a nonlinear dispersion. This has to be taken into account if correlation functions beyond the scaling limit are to be determined. Attempts to include a curved dispersion have led to the formulation of the nonlinear Luttinger liquid phenomenology [6]. But before we come to this, let us first consider the effects of a different RG irrelevant term that can be treated exactly using bosonization, namely the momentum dependence of the coupling functions $g_{2/4}(q)$.

This momentum dependence is often neglected due to its RG irrelevance, which, however, leads to an ultraviolet divergence in the (field theoretical) TLM. This becomes explicit in the computation of correlation functions and is routinely regularized "by hand" introducing a high energy cutoff (leaving the framework of constructive bosonization). We emphasize that the replacement of $g_{2/4}(q)$ by coupling constants is not required to obtain closed expressions for correlation functions; bosonization can also be applied for coupling functions. For physically resonable (screened) two-particle interactions $g_{2/4}(q)$ which decay on a scale $q_c \ll k_F$ no ultraviolet divergency occurs and an adhoc regularization can be avoided. The momentum dependence merely leads to additional momentum integrals in closed analytical expressions for the correlation functions of the TLM [7,8]. We note that momentum-dependent interactions

lead, in a first order perturbative calculation for the self-energy of the electron gas, to an effective fermion dispersion which is nonlinear. Still, the TLM with linear fermion dispersion and momentum-dependent interactions is a well-defined model that should fall into the TLL universality class.

In accordance with the RG irrelevance of the momentum dependence of the two-particle potential in generic 1d models with a metallic ground state, one can prove that the replacement of $g_{2/4}(q)$ by coupling constants and the subsequent adhoc regularization does not affect observables and correlation functions of the TLM in the scaling limit, i.e., if all energy scales are sent to zero [7,8]. However, it was shown that the physical properties are affected by the replacement if one energy scale is taken to be nonvanishing [7,8].

Consider as an example the momentum-resolved single-particle spectral function of the TLM and fix the momentum at $k - k_{\mathrm{F}} \neq 0$. Within the standard adhoc procedure [9,10], or for a box-shaped momentum dependence of the two-particle potentials [11] $g_{2/4}(q) = g_{2/4}\Theta\left(q_{\mathrm{c}}^2 - q^2\right)$, the spectral function as a function of $\omega$ shows power-law behavior at characteristic (threshold) energies. This is destroyed if any curvature of the two-particle potentials $g_{2/4}(q)$ at $q = 0$ is taken into account (that is, if generic $g_{2/4}(q)$ are considered). This result is not at odds with the RG irrelevance of the momentum dependence: the momentum $k - k_{\mathrm{F}} \neq 0$ sets a nonvanishing energy scale, but RG arguments can only be used if all scales are sent to zero. In contrast, for $k - k_{\mathrm{F}} = 0$ the momentum-resolved spectral function is universal and a power law as a function of $\omega$ is found regardless of the exact shape of $g_{2/4}(q)$ [7,8]. On general grounds the disappearance of the power law is to be expected for $k - k_{\mathrm{F}} \neq 0$. As long as $g_{2/4}(q)$ is completely flat at $q = 0$, which is explicitly the case for a box-shaped potential and implicitly assumed in the adhoc procedure [12], the system shows critical properties even for $k - k_{\mathrm{F}} \neq 0$. However, for $g_{2/4}(q)$ not being completely flat any nonvanishing $k - k_{\mathrm{F}}$ "probes" the curvature and criticality is spoiled. Mathematically, the phrase "completely flat" refers to the vanishing of the $n$-th derivative of $g_{2/4}(q)$ at $q = 0$ for all $n \in \mathbb{N}$. The analytical and numerical results of Refs. [7,8] show explicitly that the TLL universality only holds if all energy scales are sent to zero, but not for the spectral function as a function of $\omega$ at fixed $k - k_{\mathrm{F}} \neq 0$.

The discussion of the preceding paragraphs shows that effects of the (RG irrelevant) $q$-dependence of the two-particle interaction can be studied in the framework of the TLM using bosonization. Already at the time the TLL universality was introduced it was suggested to similarly investigate the (RG irrelevant) effect of the curvature of the single-particle dispersion within the TLM. This curvature leads to terms in the Hamiltonian containing three and more of the bosonic ladder operators corresponding to the elementary collective excitations. The idea is to include these perturbatively in the computation of observables and correlation functions [3]. However, the attempts in this direction led to divergences which indicate a breakdown of the corresponding perturbation theory [13–16]. The insights gained along this route are thus rather limited. This prompted alternative attempts to study the influence of the curvature of the single-particle dispersion on spectral functions at nonvanishing momenta.

Taken that the elementary excitations of the TLM are associated to the (bosonic) density operators $\rho_q^\dagger$ [1–4], the most natural spectral function to consider is the dynamical structure factor (DSF)

$$S(q,\omega) = \frac{2\pi}{L} \sum_n \left| \langle E_n | \rho_q^\dagger | E_0 \rangle \right|^2 \delta\left(\omega - [E_n - E_0]\right) \tag{1}$$

at a small momentum $0 < q < q_{\mathrm{c}}$ [6]. Here, $|E_n\rangle$ denotes the many-body eigenstates, $E_n$ the

corresponding energies, and $L$ is the system size. For the noninteracting spinless electron gas with quadratic dispersion it can be easily computed and for $\omega > 0$ is given by ($L \to \infty$)

$$S_0(q,\omega) = \frac{m}{q} \Theta\left[\omega - \omega_-(q)\right] \Theta\left[\omega_+(q) - \omega\right], \tag{2}$$

where

$$\omega_\pm(q) = v_{\mathrm{F}} q \pm q^2/(2m) \tag{3}$$

denotes the threshold values, $m$ the fermion mass, and $v_{\mathrm{F}}$ the Fermi velocity. Likewise, the DSF can be calculated analytically for the TLM, where for $\omega > 0$ and $L \to \infty$

$$S_{\mathrm{TL}}(q,\omega) = q \frac{1 + \frac{g_4(q)}{2\pi v_{\mathrm{F}}} - \frac{g_2(q)}{2\pi v_{\mathrm{F}}}}{\sqrt{\left[1 + \frac{g_4(q)}{2\pi v_{\mathrm{F}}}\right]^2 - \left[\frac{g_2(q)}{2\pi v_{\mathrm{F}}}\right]^2}} \delta\left[\omega - \omega(q)\right]. \tag{4}$$

Here,

$$\omega(q) = v_{\mathrm{F}} q \sqrt{\left[1 + \frac{g_4(q)}{2\pi v_{\mathrm{F}}}\right]^2 - \left[\frac{g_2(q)}{2\pi v_{\mathrm{F}}}\right]^2} \tag{5}$$

is the dispersion of the TLM bosons. Comparing $S_{\mathrm{TL}}(q,\omega)$ for $g_{2/4}(q) = 0$ with $S_0(q,\omega)$, the consequence of the linearization becomes obvious. The box-like shape at fixed $q$ degenerates to a $\delta$-peak.

It is straightforward to compute the correction $S_1(q,\omega)$ to $S_0(q,\omega)$ to first order in a two-particle potential $V(q)$ [see the diagrams (a) and (b) of Fig. 1] [6, 17]. Details of the computation are given in Sect. 2. Close to but above the lower threshold value $\omega_-(q)$ the leading behavior of the correction is given by

$$S_1(q,\omega) \sim \frac{m}{q} \frac{m}{\pi q} \left[V(q) - V(0)\right] \ln\left[\frac{\omega - \omega_-(q)}{\delta\omega(q)}\right], \tag{6}$$

with

$$\delta\omega(q) = \omega_+(q) - \omega_-(q) = \frac{q^2}{m}. \tag{7}$$

This result was interpreted as the linear term in an expansion in powers of $\alpha(q) \ln\left[\frac{\omega - \omega_-(q)}{\delta\omega(q)}\right]$ with the properly resummed result being a power law with exponent $\alpha(q)$ for $\omega \searrow \omega_-(q)$ (exponential series) [17].

An analogy to the exactly solvable Fermi edge singularity problem [18] was drawn. Mahan first analyzed the spectral function of this problem using perturbation theory in the interaction [19]. He showed that the $\ln^n$-terms up to third order are consistent with a power-law threshold singularity, and based on this he conjectured this form. This conjecture was confirmed shortly after by comparison to the exact solution; see Ref. [18] and references therein. Referring to his perturbative calculation Mahan noted that "Of course, one cannot guarantee that the series [...] is an exponential without evaluating the series to all orders." [19]. In the light of this it is surprising that already the first order result Eq. (6) was considered to be sufficient to conjecture a similar power-law threshold singularity at $\omega_-(q)$ with exponent

$$\alpha(q) = -\frac{m}{\pi q} \left[V(0) - V(q)\right] \tag{8}$$

in the DSF of the 1d interacting electron gas [6, 17]. A field theory, the so-called mobile impurity model, constructed in such a way that it leads to this power law, was argued to be the appropriate effective model [6, 17]. In a first step, we here provide the calculation of the second order correction to the DSF of the interacting electron gas. Our result shows that the perturbative expansion is indeed consistent with a power law with exponent Eq. (8) up to second order. As emphasized by Mahan, one has to go to infinite order to prove a power law. Nevertheless, a confirmation up to second order at least strengthens the conjecture.

Despite this consistency with the prediction of the mobile impurity model, in a second step, we take this insight as a motivation to critically assess the steps leading to this model. It was used to not only compute the DSF, but also to evaluate other correlation functions such as, e.g., the single-particle spectral function [6]. Within the mobile impurity model, the calculation of those is straightforward [20, 21] and results in power laws with momentum dependent exponents. However, this is not the focus of our argument below. Rather, we are concerned with the construction of the mobile impurity model itself. Several of the crucial steps rely on heuristic arguments [6, 22]. In particular, we show that it is impossible to include the momentum dependence of the (bulk) two-particle potential without sacrificing the possibility to exactly solve the mobile impurity model. As discussed above, this RG irrelevant momentum dependence was shown to spoil power-law scaling of the single-particle spectral function at $k - k_{\mathrm{F}} \neq 0$ which was widely believed to be part of the TLL universality [7, 8]. Even in the light of our finding that the second order perturbation theory for the DSF is consistent with a power-law behavior at the lower threshold this raises doubts that the mobile impurity model can really be considered as the basis of a new type of universality, namely the nonlinear Luttinger liquid phenomenology [6]. We conclude that more work is required to clarify this.

The rest of this article is structured as follows. In Sect. 2 we introduce the model and provide details of the first order calculation of the DSF. Section 3 is devoted to our second order calculation. Details of the computations of Sect. 3 are presented in the Appendix. In Sect. 4 we show that including the momentum dependence of the (bulk) two-particle potential spoils the exact solvability of the mobile impurity model. We discuss the implications of our findings in Sect. 5. In this we also briefly describe earlier numerical [22–25] and analytical [26, 27] results obtained for models other than the electron gas. In particular, we mention recent analytical insights [28, 29] gained for the XXZ Heisenberg model, which is equivalent to a model of spinless fermions with nearest-neighbor interaction and hopping. It falls into the nongeneric class of integrable models with short-ranged interactions. For this special model, the analytical results support the nonlinear Luttinger liquid phenomenology.

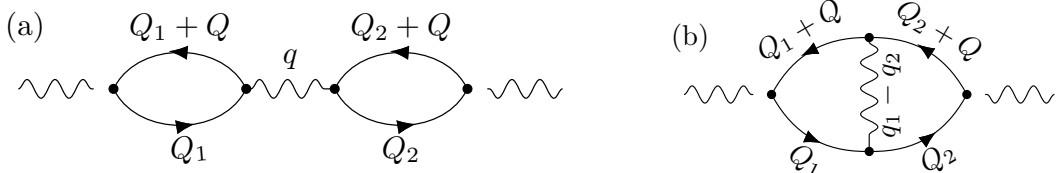

Figure 1: Feynman diagrams for the polarization in first order in the interaction. The capital labels are a multi-index, e.g. $Q_1 + Q = (q_1 + q, \mathrm{i}[\omega_1 + \omega])$. Internal variables are summed over.

## 2  The model and first order perturbation theory

We consider interacting spinless fermions with quadratic dispersion on a ring of length $L$, with the Hamiltonian

$$H = H_0 + H_{\mathrm{int}}, \tag{9}$$

$$H_0 = \sum_k \xi_k c_k^\dagger c_k, \quad \xi_k = \frac{k^2}{2m} - \frac{k_{\mathrm{F}}^2}{2m} \tag{10}$$

$$H_{\mathrm{int}} = \frac{1}{2L} \sum_{k_1, k_2, k_3} V(k_1) c_{k_2}^\dagger c_{k_3}^\dagger c_{k_3 + k_1} c_{k_2 - k_1}. \tag{11}$$

As mentioned previously, we assume that the (screened) interaction potential $V(q)$ vanishes on a scale $q_{\mathrm{c}}$. Besides, as physically reasonable, $V(q)$ should be a smooth, even function.

Let us briefly recapitulate the calculation of the DSF of the noninteracting system and the first order correction of the interacting system. At zero temperature and for $\omega > 0$, $S(q, \omega)$ can be related to the imaginary part of the polarization via the fluctuation-dissipation theorem,

$$S(q, \omega) = -2 \, \mathrm{Im}\{\chi(\mathrm{q}, \omega)\}. \tag{12}$$

Diagrammatically, the polarization for the noninteracting system is simply given by the electron bubble. An analytic evaluation gives

$$\chi_0(q, \omega) = \frac{m}{2\pi q} \ln \left| \frac{\omega^2 - \omega_-(q)^2}{\omega^2 - \omega_+(q)^2} \right| - \mathrm{i} \frac{m}{2q} \Theta \left[ \omega - \omega_-(q) \right] \Theta \left[ \omega_+(q) - \omega \right], \tag{13}$$

from which the noninteracting DSF Eq. (2) immediately follows.

The two relevant Feynman diagrams in first order in the interaction are depicted in Fig. 1. We do not consider the self-energy corrections. They contribute to a shift in the threshold energy, and such a shift does not produce logarithmic corrections to the DSF [see Eq. (18) below]. Straight lines with arrows denote free Green's functions $G_0(\mathrm{i}\omega, q) = 1/(\mathrm{i}\omega - \xi_q)$, and the wiggled lines the interaction. Note that the wiggled lines to the left and the right of the diagram are actually amputated, and only drawn here to make frequency and momentum conservation explicit. We do not write the conventional minus-sign in front of the diagrams. The RPA-like diagram Fig. 1 (a) can be easily computed from the result for the polarization bubble, and we obtain for the DSF

$$\begin{aligned} S_1^{\mathrm{a}}(q, \omega) &= -2 \, \mathrm{Im} \left\{ \mathrm{V}(\mathrm{q}) \left[ \chi_0(\mathrm{q}, \omega) \right]^2 \right\} \\ &= \frac{m^2}{\pi q^2} V(q) \ln \left| \frac{\omega^2 - \omega_-(q)^2}{\omega^2 - \omega_+(q)^2} \right| \Theta \left[ \omega - \omega_-(q) \right] \Theta \left[ \omega_+(q) - \omega \right]. \end{aligned} \tag{14}$$

For $\omega \searrow \omega_-(q)$, this behaves as

$$S_1^{\mathrm{a}}(q,\omega) \sim \frac{m^2}{\pi q^2} V(q) \ln\left[\frac{\omega - \omega_-(q)}{\delta\omega(q)}\right]. \tag{15}$$

The calculation of the vertex correction $S_1^{\mathrm{b}}(q,\omega)$ is not much more difficult; after performing the Matsubara summations and going to $L \to \infty$ as well as $\mathrm{i}\omega \to \omega + \mathrm{i}\eta$, we obtain

$$\chi_1^{\mathrm{b}}(q,\omega) = -\int \frac{\mathrm{d}q_1 \mathrm{d}q_2}{(2\pi)^2} V(q_2 - q_1) \prod_{j=1}^{2} \frac{\Theta[k_{\mathrm{F}}^2 - q_j^2] - \Theta[k_{\mathrm{F}}^2 - (q_j + q)^2]}{\omega + \mathrm{i}\eta - \frac{q^2}{2m} - \frac{qq_j}{m}}. \tag{16}$$

Since we are only interested in the imaginary part of this expression, we can use the Sokhotski-Plemelj theorem to integrate out one variable analytically [30]. The remaining expression can be analyzed using integration by parts, where the term including derivatives of the interaction potential is not relevant in this context as it produces only subleading corrections. The leading behavior for $\omega$ close to $\omega_-(q)$ is given by

$$S_1^{\mathrm{b}}(q,\omega) \sim -\frac{m^2}{\pi q^2} V(0) \ln\left[\frac{\omega - \omega_-(q)}{\delta\omega(q)}\right]. \tag{17}$$

Together with the correction from the RPA-like diagram Eq. (15), this gives the first order result Eq. (6).

Let us now consider what an expansion of a power law with an interaction dependent exponent looks like. We denote a shifted threshold by $\tilde{\omega} = \omega_0 + \omega_1 v + \dots$ with a small parameter $v$, which in our context is given by the interaction strength, and the exponent by $\alpha = \alpha_1 v + \alpha_2 v^2 + \dots$. Then,

$$
\begin{aligned}
(\omega - \tilde{\omega})^\alpha =& 1 + \alpha_1 \ln(\omega - \omega_0)\, v \\
&+ \left[-\frac{\alpha_1 \omega_1}{\omega - \omega_0} + \alpha_2 \ln(\omega - \omega_0) + \frac{1}{2}\alpha_1^2 \ln^2(\omega - \omega_0)\right] v^2 + \mathcal{O}(v^3).
\end{aligned}
\tag{18}
$$

As already discussed, the zeroth and first order term in a perturbative calculation of $S(q,\omega)$, Eqs. (2) plus (6), are consistent with such an expansion with the threshold $\omega_-(q)$ and the exponent $\alpha(q)$.

## 3  Second order perturbation theory

Next, we compute the second order correction to the DSF for the interacting electron gas. For consistency with a power law, we read off from Eq. (18) that the leading logarithmically divergent term would have to have the prefactor $\alpha^2(q)/2$ in front of a logarithm squared. Below, we will thus only consider the leading divergent contributions.

As it is more efficient, for the second order calculation we use Hugenholtz diagrams, where a circle represents the antisymmetrized interaction vertex. Again, we only consider the diagrams that do not contain self-energy insertions, as those are expected to contribute to a shift in the energy threshold which does not yield logarithmic corrections [cf. Eq. (18)]. We thus have to examine the three diagrams shown in Fig. 2.

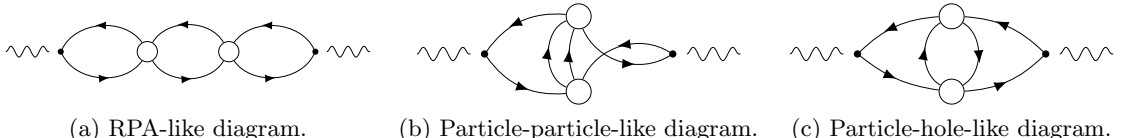

(a) RPA-like diagram.     (b) Particle-particle-like diagram.     (c) Particle-hole-like diagram.

Figure 2: Second order Hugenholtz diagrams for the polarization without self-energy corrections.

The formula for the RPA-like diagram shown in Fig. 2 (a) is given by

$$\chi_2^{\mathrm{a}}(q,\omega) = \int \frac{\mathrm{d}q_1 \mathrm{d}q_2 \mathrm{d}q_3}{(2\pi)^3} \left[V(q) - V(q_1 - q_2)\right]\left[V(q) - V(q_2 - q_3)\right]$$
$$\times \prod_{j=1}^{3} \frac{\Theta[k_{\mathrm{F}}^2 - q_j^2] - \Theta[k_{\mathrm{F}}^2 - (q_j + q)^2]}{\omega + \mathrm{i}\eta - \frac{q^2}{2m} - \frac{qq_j}{m}}, \tag{19}$$

after performing the Matsubara summation, taking $L \to \infty$, and analytic continuation $\mathrm{i}\omega \to \omega + \mathrm{i}\eta$. Evaluating the intergrals and extracting the leading divergence requires steps similar to the ones taken to compute the first order RPA-like diagram $\chi_1^{\mathrm{b}}(q,\omega)$ of Eq. (16). The leading behavior close to the lower threshold is

$$S_2^{\mathrm{a}}(q,\omega) \sim \frac{m}{q}\frac{3}{4}\alpha^2(q)\ln^2\left[\frac{\omega - \omega_-(q)}{\delta\omega(q)}\right]. \tag{20}$$

The other diagrams Fig. 2 (b) and 2 (c) are much more difficult to evaluate. Here, we therefore do not even give the analytic expressions of those in terms of integrals and only present the final leading behavior. The integral expressions as well as the main steps of their evaluation are given in the Appendix. The particle-particle-like diagram depicted in Fig. 2 (b) contributes close to $\omega_-(q)$ as

$$S_2^{\mathrm{b}}(q,\omega) \sim -\frac{m}{q}\frac{1}{4}\alpha^2(q)\ln^2\left[\frac{\omega - \omega_-(q)}{\delta\omega(q)}\right]. \tag{21}$$

The particle-hole-like diagram, Fig. 2 (c), gives only subleading contributions. Note that for this diagram, a very careful evaluation is necessary to ensure that it is finite away from $\omega = \omega_\pm(q)$. Details can be found in the Appendix.

Taking the results Eqs. (20) and (21) together, we obtain for the leading behavior for $\omega$ close to but above $\omega_-(q)$

$$S_2(q,\omega) \sim \frac{1}{2}\alpha^2(q)\ln^2\left[\frac{\omega - \omega_-(q)}{\delta\omega(q)}\right], \tag{22}$$

which is consistent with a power law. As emphasized in the Introduction this consistency does not prove the appearance of a power law with exponent Eq. (8) close to the lower threshold of the DSF, but it at least strengthens the confidence in the conjectured behavior [6,17]. The mobile impurity model was introduced as the effective model to capture this power law and used to compute other correlation functions, e.g., the single-particle spectral function. It was argued to form the basis of the entire nonlinear Luttinger liquid phenomenology. Despite the consistency of the second order perturbative result with the conjectured power law, we next critically assess one of the crucial the steps leading to this model.

# 4 Towards the mobile impurity model

The mobile impurity model Hamiltonian was constructed from the interacting electron gas Hamiltonian Eq. (9) for weak interactions [6, 17]. For this, as a first step, the creation and annihilation operators in momentum space are projected on three narrow subbands around $k_F$ (right movers), $-k_F$ (left movers) and $k_F - q$ (deep hole/impurity, $0 < q < q_c$),

$$c_k \longrightarrow \underbrace{c_{R,k-k_F}}_{k \approx k_F} + \underbrace{c_{L,k+k_F}}_{k \approx -k_F} + \underbrace{d_{k-(k_F-q)}}_{k \approx (k_F - q)}. \tag{23}$$

The operators on the right-hand side are only nonzero for momenta close to the one indicated in the underbrace. Projecting the operators in the kinetic energy Eq. (10) and subsequently linearizing the dispersion in each subband, one obtains

$$H_0 \to \sum_{k \, (k \, \text{small})} \left[ v_F k \left( c_{R,k}^\dagger c_{R,k} - c_{L,k}^\dagger c_{L,k} \right) + (\xi_{k_F - q} + v_d k) \, d_k^\dagger d_k \right], \tag{24}$$

where we have defined $v_d = (k_F - q)/m$. The linearization is performed keeping in mind that the three subbands have only a small width. This projection and linearization can only be justified heuristically [22]. In particular, there is, no rigorous argument for this related to the RG irrelevance of additional terms or similar. The energy scales involved and the resulting crossover to the results obtained within the Tomonaga-Luttinger model have been discussed for the single-particle spectral function in Ref. [31]; see also the review [6]. However, these papers ignore that for this spectral function and $k \neq k_F$ there is no universal Tomonaga-Luttinger liquid result avaliable. This was shown in Refs. [7] and [8].

Concerning the interacting part of the Hamiltonian, Eq. (11), one quickly sees that the momentum transfer via the potential–$k_1$ in Eq. (11)–can only be close to the momenta $0, \pm q, \pm 2k_F$, and $\pm(2k_F - q)$ due to kinematic constraints. For $k_1$ close to zero the interaction can straightforwardly be written as a quadratic form in the densities of the right/left movers and the deep hole (considering only a single deep hole) [6,22]. This term can be treated exactly using bosonization. As usual, we assume that $V(k)$ decays quickly for $|k| > q_c$, and that $0 < q < q_c \ll k_F$. Therefore, we do not consider the terms stemming from $H_{int}$ with $k_1$ close to $\pm 2k_F, \pm(2k_F - q)$. If desired, they can be taken into account by partially neglecting the momentum dependence of the interaction potential in analogy to the term with $k_1$ close to $\pm q$ to be discussed next. For this we obtain after projection

$$-\frac{1}{L} \sum_{\substack{k_1', k_2', k_3' \\ (k_i' \, \text{small})}} V(q - k_1' + k_2' - k_3') \, c_{R,k_2'}^\dagger c_{R,k_2' - k_1'} \, d_{k_3'}^\dagger d_{k_3' + k_1'}. \tag{25}$$

The standard procedure [6, 22] is to partially neglect the momentum dependence of $V$ by setting $V(q - k_1' + k_2' - k_3') \to V(q)$, as all $k_i'$ are much smaller than $q$, which allows to take the potential out of the sum. However, for the reasons given in the Introduction, we are interested in keeping the full momentum dependence of the interaction potential. Unfortunately, for $k_1 \approx \pm q$ [remember that $k_1$ refers to the $k_1$ of Eq. (11)] this is not possible due to the following problem. In order to solve the mobile impurity model Hamiltonian, a unitary transform to a noninteracting Hamiltonian is used [6]. For this, it is crucial that the mobile impurity Hamiltonian only contains the densities in the interaction part. In Eq. (25), the dependence

of $V$ on $k_2$ and $k_3$ prevents us from writing it in terms of the densities. We therefore have to neglect this dependence because we would otherwise spoil the exact solvability of the mobile impurity Hamiltonian. We also attempted to use a Taylor expansion in $k_2$ and $k_3$, but the details of this are beyond the scope of this work. To summarize, we have neither been able to find a way to keep the full momentum dependence of $V$ nor found a justification for partially neglecting it. We emphasize that this is in contrast to the derivation of the TLM, where the approximations are legitimated by RG arguments. One can only proceed by purely pragmatically replacing Eq. (25) by

$$-\frac{1}{L} \sum_{k\,(k\,\text{small})} \frac{V(q-k)+V(q+k)}{2}\, \rho_{\mathrm{R},-k}\rho_{\mathrm{d},k}. \tag{26}$$

Note that the structure with $[V(q-k)+V(q+k)]/2$ is necessary to retain a Hermitian Hamiltonian. This results in the mobile impurity Hamiltonian which can be solved by bosonization and a unitary transformation. We stress that it is not possible, starting from the interacting electron gas, to arrive at the mobile impurity model employing only controlled approximations.

## 5   Discussion

As already emphasized, computing the DSF within the mobile impurity model leads to a power law at the lower threshold with an exponent which, to leading order in the two-particle interaction, agrees with Eq. (8) [6, 17]. We found consistency with this behavior within second order perturbation theory (in the two-particle potential) for the interacting electron gas. Next to the DSF, also other observables such as, e.g., the single-particle spectral function of the mobile impurity model were computed [6]. The spectral function shows power-law threshold singularities with momentum-dependent exponents as well. This power-law behavior of correlation functions was interpreted as a new type of universality which was embraced in the nonlinear Luttinger liquid phenomenology [6]. As the foundation of the mobile impurity Hamiltonian is rather heuristic [22] it is not clear whether the correlation functions of the mobile impurity model are indeed equivalent to those of, e.g., the interacting electron gas. In particular, as discussed in the last section, in the construction of the mobile impurity model the momentum dependence of the two-particle interaction is kept only partly. Neglecting this momentum dependence led to spurious power laws at $k-k_{\mathrm{F}} \neq 0$ in the single-particle spectral function of the TLM [7, 8] and a corresponding alleged universality. It cannot be excluded that something similar happens in the presence of band curvature. In the mobile impurity Hamiltonian only the fermionic densities appear, which is crucial for the exact solvability. The DSF is the double Fourier transform of the density-density correlation function in $x$ and $t$. In contrast, the single-particle spectral function is the double Fourier transform of the correlation function of the field operators. Those do not appear in the mobile impurity model by construction. The DSF might therefore be a special case. A more fundamental open question in this respect concerns the mechanism underlying the conjectured universality. It must be completely different from the one being at the heart of TLL theory (quantum critical behavior, scaling and conformal invariance).

Two attempts to further substantiate the nonlinear Luttinger liquid phenomenology of 1d fermionic many-body systems are the numerical computation of correlation functions for lattice models [22–25] and analytical studies of exactly solvable (integrable) models [26–29].

Both approaches can even be combined when it comes to the numerical evaluation of matrix elements for Bethe ansatz solvable models [24].

Some of the results obtained employing numerical methods were interpreted to be consistent with the nonlinear Luttinger liquid phenomenology. However, they suffer from the crucial shortcoming of a rather small energy resolution due to finite size effects [23–25]. This severly limits the possibilty to convincingly demonstrate power-law scaling.

The DSF [26] and the single-particle spectral function [27] of the Calogero-Sutherland model were computed analytically employing integrability. However, this model is characterized by a long-ranged two-particle interaction and does thus not fulfill the criteria of the nonlinear Luttinger liquid phenomenology; it does not fall into the proper class of models [6].

There is one special case in which the predictions of the nonlinear Luttinger liquid phenomenology were confirmed in an analytical analysis. In Ref. [28], the singular behavior of the dynamical response function of the integrable XXZ Heisenberg model in the gapless regime, which is equivalent to the lattice model of spinless fermions with nearest-neighbor hopping and interaction in its metallic phase, was studied. The analysis exploits integrability and builds on the form factor expansion [29] but "does not rely, at any stage, on some hypothetical correspondence with a field theory or other phenomenological approaches" [28]. For the specific integrable model with short-ranged interaction the results for the threshold power laws and exponents are in agreement with the ones obtained within the nonlinear Luttinger liquid phenomenology. Reference [28] also provided arguments that this should be valid for other integrable models as well.

We conclude that more research is required to verify the predictions obtained with the mobile impurity model, in particular, for correlation functions other than the dynamical structure factor and generic (nonintegrable) models [32]. Put differently, the effect of band curvature on correlation functions of generic 1d interacting Fermi systems beyond the low-energy scaling limit (in which the curvature is RG irrelevant) remains an open issue which deserves further investigations. Based on the arguments presented here we believe that it is unlikely that this will lead to any "finite energy" universal theory applicable to a broad class of models in analogy to the TLL theory, which holds if all energy scales are sent to zero. One step along the lines of the present work would be to compute the single-particle spectral function of the 1d electron gas in second order perturbation theory in the two-particle interaction. We leave this for the future.

# Acknowledgements

We are grateful to Kurt Schönhammer, Imke Schneider, Patrick Plötz, Dirk Schuricht, Vladimir Gritsev, Jean-Sébastien Caux, Sasha Gamayun, and Frank Göhmann for discussions.

**Funding information**   This work was supported by the Deutsche Forschungsgemeinschaft via RTG 1995 (LM, MP, VM).

# Appendix: Second order calculation

We here provide details on the second order perturbative correction to the polarization. For brevity, we set the mass to $m = 1$ and only restore this after the calculation. We start with the diagram depicted in Fig. 2 (b). In the thermodynamic limit, after analytic continuation, we find

$$\frac{1}{2} \int \frac{\mathrm{d}q_1 \mathrm{d}q_2 \mathrm{d}q_3}{(2\pi)^3} \left( \Theta[k_\mathrm{F}^2 - q_3^2]\Theta[q_1^2 - k_\mathrm{F}^2] + \Theta[q_3^2 - k_\mathrm{F}^2]\Theta[k_\mathrm{F}^2 - q_1^2] \right) \Theta[k_\mathrm{F}^2 - q_2^2]$$

$$\times \left\{ \frac{[V(q_2 - q_1) - V(q_3 - q_1)][V(q_3 - q_1 - q) - V(q_2 - q_1 - q)]}{(\omega + \mathrm{i}\eta - \frac{q^2}{2} - qq_1)(q_2 - q_1)(q_1 - q_3)(\omega + \mathrm{i}\eta - \xi_{q_3} + \xi_{q_1} + \xi_{q_2 - q_1 + q_3 - q} - \xi_{q_2})} \right.$$

$$- \frac{[V(q_2 - q_3) - V(q_3 - q_1)][V(q_3 - q_1 - q) - V(q_2 - q_3 - q)]}{(\omega + \mathrm{i}\eta - \frac{q^2}{2} - qq_1)(\omega + \mathrm{i}\eta + \frac{q^2}{2} - qq_2)(q_2 - q_3)(q_1 - q_3)}$$

$$- \frac{[V(q_2 - q_3 + q) - V(q_3 - q_1)][V(q_3 - q_1 - q) - V(q_2 - q_3)]}{(\omega + \mathrm{i}\eta - \frac{q^2}{2} - qq_1)(\omega + \mathrm{i}\eta - \frac{q^2}{2} - qq_2)(\omega + \mathrm{i}\eta - \xi_{q_3} + \xi_{q_1} + \xi_{q_2} - \xi_{q_2 - q_3 + q_1 + q})}$$

$$+ \frac{[V(q_2 - q_1 + q) - V(q_3 - q_1 + q)][V(q_3 - q_1) - V(q_2 - q_1)]}{(\omega + \mathrm{i}\eta + \frac{q^2}{2} - qq_1)(\omega + \mathrm{i}\eta + \xi_{q_3} - \xi_{q_1} - \xi_{q_2 - q_1 + q_3 + q} + \xi_{q_2})(q_1 - q_3)(q_2 - q_1)}$$

$$- \frac{[V(q_2 - q_3 + q) - V(q_3 - q_1 + q)][V(q_3 - q_1) - V(q_2 - q_3)]}{(\omega + \mathrm{i}\eta + \frac{q^2}{2} - qq_1)(\omega + \mathrm{i}\eta - \frac{q^2}{2} - qq_2)(q_1 - q_3)(q_2 - q_3)}$$

$$\left. + \frac{[V(q_2 - q_3) - V(q_3 - q_1 + q)][V(q_3 - q_1) - V(q_2 - q_3 - q)]}{(\omega + \mathrm{i}\eta + \frac{q^2}{2} - qq_1)(\omega + \mathrm{i}\eta + \frac{q^2}{2} - qq_2)(\omega + \mathrm{i}\eta + \xi_{q_3} - \xi_{q_1} - \xi_{q_2} + \xi_{q_2 - q_3 + q_1 - q})} \right\}. \tag{27}$$

The denominators of the form $\omega + \mathrm{i}\eta + \dots$ can be used to eliminate one integration variable by using the Sokhotski-Plemelj theorem when taking the imaginary part of this expression. The arising delta-functions are easy to evaluate in the case of the denominators of the form $\omega + \mathrm{i}\eta \pm q^2/2 - qq_j$. They result in the contribution given in Eq. (21).

For the more complicated denominators, the calculation is more involved. Take, e.g., the term $\omega + \mathrm{i}\eta - \xi_{q_3} + \xi_{q_1} + \xi_{q_2 - q_1 + q_3 - q} - \xi_{q_2}$. In this case, it is not immediately obvious which variable should be eliminated with the delta-function. Since the term is quadratic in $q_1$, it might seem favorable to eliminate $q_2$ or $q_3$. But then, the momentum argument of the potential is given by a nonlinear function of $q_1$ and $q_{2/3}$, and the resulting terms are very difficult to correctly evaluate further. Specifically, in the treatment with the integration by parts the derivative of the potential can no longer be neglected. A straightforward way to see this is to first eliminate $q_2$, treat the arising integrals as before (neglecting the derivatives of $V$) and then compare to the analogous calculation where $q_3$ has been eliminated first. The results do not agree, which illustrates that this procedure is incorrect. Instead, suitable shifts of the variables make it possible to eliminate $q_1$ first, and the integrand then takes the form

$$\frac{1}{8\pi^2 q}[V(q_2) - V(q_3)][V(q_3 - q) - V(q_2 - q)]$$

$$\times \frac{1}{(q_2 - q)(q_3 - q)q_2 q_3}. \tag{28}$$

An analysis of the integrals in this formulation shows that there are no contributions of the form $\ln^2 |[\omega - \omega_\pm(q)]/\delta\omega(q)|$.

We continue with the particle-hole-like diagram shown in Fig. 2 (c). We have to evaluate

$$
2 \int \frac{\mathrm{d}q_1 \mathrm{d}q_2 \mathrm{d}q_3}{(2\pi)^3} \Theta[q_1^2 - k_{\mathrm{F}}^2] \Theta[k_{\mathrm{F}}^2 - q_2^2] \Theta[k_{\mathrm{F}}^2 - q_3^2]
$$

$$
\times \left\{ \frac{[V(q_2 - q_1) - V(q_3 - q_1)][V(q_2 - q_1) - V(q_3 - q_1 - q)]}{(\omega + \mathrm{i}\eta - \frac{q^2}{2} - qq_1)(\omega + \mathrm{i}\eta - \frac{q^2}{2} - qq_2)(q_1 - q_2)(q_3 - q_1)} \right.
$$

$$
+ \frac{[V(q_2 - q_1 - q) - V(q_3 - q_1)][V(q_2 - q_1 - q) - V(q_3 - q_1 - q)]}{(\omega + \mathrm{i}\eta - \frac{q^2}{2} - qq_1)(\omega + \mathrm{i}\eta + \frac{q^2}{2} - qq_2)(\omega + \mathrm{i}\eta - \xi_{q_3} + \xi_{q_1} - \xi_{q_2} + \xi_{q_2 + q_3 - q_1 - q})}
$$

$$
- \frac{[V(q_2 - q_3) - V(q_3 - q_1)][V(q_2 - q_3) - V(q_3 - q_1 - q)]}{(\omega + \mathrm{i}\eta - \frac{q^2}{2} - qq_1)(q_3 - q_2)(q_3 - q_1)(\omega + \mathrm{i}\eta - \xi_{q_3} + \xi_{q_1} - \xi_{q_2 - q_3 + q_1 + q} + \xi_{q_2})} \tag{29}
$$

$$
- \frac{[V(q_2 - q_1 + q) - V(q_3 - q_1 + q)][V(q_2 - q_1 + q) - V(q_3 - q_1)]}{(\omega + \mathrm{i}\eta + \frac{q^2}{2} - qq_1)(\omega + \mathrm{i}\eta - \frac{q^2}{2} - qq_2)(\omega + \mathrm{i}\eta + \xi_{q_3} - \xi_{q_1} + \xi_{q_2} - \xi_{q_2 + q_3 - q_1 + q})}
$$

$$
+ \frac{[V(q_2 - q_1) - V(q_3 - q_1 + q)][V(q_2 - q_1) - V(q_3 - q_1)]}{(\omega + \mathrm{i}\eta + \frac{q^2}{2} - qq_1)(\omega + \mathrm{i}\eta + \frac{q^2}{2} - qq_2)(q_1 - q_2)(q_3 - q_1)}
$$

$$
\left. - \frac{[V(q_2 - q_3) - V(q_3 - q_1 + q)][V(q_2 - q_3) - V(q_3 - q_1)]}{(\omega + \mathrm{i}\eta + \frac{q^2}{2} - qq_1)(q_3 - q_2)(q_3 - q_1)(\omega + \mathrm{i}\eta + \xi_{q_3} - \xi_{q_1} + \xi_{q_2 - q_3 + q_1 - q} - \xi_{q_2})} \right\}.
$$

Using the Sokhotski-Plemelj theorem to obtain the imaginary part of this, we find that the integrals can all be brought to a form with the integrand

$$
\frac{1}{4\pi^2 q^2} [V(q_1) - V(q_3)][V(q_1) - V(q_3 - q)] \frac{1}{q_1^2 q_3}. \tag{30}
$$

We note that the integration regions touch the singular lines $q_1 = 0$ and $q_3 = 0$ in several points. However, the arising integrals that are singular everywhere [not only for $\omega = \omega_\pm(q)$] exactly cancel out. An analytic evaluation then shows that this diagram produces only subleading contributions.

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
