# Peer review of "Investigating the roots of the nonlinear Luttinger liquid phenomenology"

_SciPost Physics_

## Round 1 · Referee Report · Anonymous (Referee 1) · 2019-4-21

Strengths

Brute-force calculation of the second-order in interaction contribution to the spectral function of 1D Fermi gas presented in Appendix.

Weaknesses

  1. No new results.
  2. Vague and misleading conclusions based on the lack of understanding of a textbook-level theory of Fermi edge singularity.

Report

This work addresses the derivation of the power-law singularity of a spectral function of interacting one-dimensional Fermi gas associated with the threshold of the spectral continuum. The Authors take the perturbative approach to the problem, and successfully reproduce the well-known lowest-order in interaction result, obtaining the $\ln\omega$ term in the asymptote (energy $\omega$ is measured from the edge). Then they proceed to the second order in the interaction potential, and -- predictably -- find the $\ln^2\omega$ contribution in the leading logarithmic approximation series. The sub-leading terms are relegated to the Appendix. The Authors state that two first terms of the leading-logarithmic expansion support the power-law form of the edge singularity. There is nothing new in this statement. Since the 1969 work of Schotte and Schotte, there are better ways to derive the asymptotic power law. Later on, Don Hamann (Phys. Rev. Lett. v. 26, 1030, 1971) offered a workable way of finding the sub-leading corrections. The Authors may want to familiarize themselves with an exposition of this problem, e.g., in the monograph by Gogolin et al, "Bosonization and Strongly Correlated Systems".

Probably the Authors could assess the corrections to the power-law asymptote using the material of their Appendix. Being sub-leading, these corrections may become important at larger $\omega$, but should not affect the validity of the power-law asymptote. Instead, the Authors added Section 4-6 which have unclear meaning and no real conclusion. The vague sense of these sections of their work is condensed into a misleading statement occupying three last sentences of the abstract to their manuscript.

Requested changes

none

  • validity: poor
  • significance: poor
  • originality: poor
  • clarity: poor
  • formatting: perfect
  • grammar: good

Author:  Lisa Markhof  on 2019-04-29  [id 504]

(in reply to Report 1 on 2019-04-21)
Category:
reply to objection

There seems to be a misunderstanding about the purpose of our work. We agree with the Referee that if it concerned the Fermi edge singularity, it would indeed be textbook material. However, we consider the problem of 1d correlated fermions with parabolic dispersion. The calculation of the second order perturbative correction to the dynamical structure factor (DSF) is therefore a difficult task and it was previously not clear - and not at all obvious - that the leading terms of this correction are indeed consistent with a power law. We stress that this is an important and new result.

In previous works (see references in our paper), an analogy between the problem at hand (the calculation of correlation functions of the 1d interacting Fermi gas) to the Fermi edge singularity problem was drawn. But in contrast to the latter, where an exact solution is known, no such exact solution confirms power laws in the DSF of the interacting Fermi gas. As we point out in our paper, the mobile-impurity model constructed to exhibit this power law can be derived only heuristically. A thorough investigation of this construction is the motivation for our Sects. 4-5. We disagree with the Referee that Sects. 4-6 are vague. We clearly state that the momentum dependence of the interaction potential, which is known to be non-negligible in the Tomonaga-Luttinger model beyond the scaling limit (see the Introduction of our paper), has to be neglected at least partially in the construction of an exactly solvable mobile-impurity model. This connection is also highlighted in the last three sentences of the abstract.

Concerning the sub-leading corrections, we wish to underline that we are only interested in the leading behavior close to the thresholds (as also clearly stated in our work). We note that we are familiar with the references the Referee points out, and we emphasize that those are about the Fermi edge singularity problem rather than about the 1d interacting electron gas. Even for the mobile-impurity model, a more complicated calculation is needed as a mobile impurity has to be considered rather than an immobile core hole.

---

## Round 1 · Referee Report · Anonymous (Referee 2) · 2019-5-31

Strengths

  1. Explicit analysis of a tricky second-order contribution to the dynamic structure factor.
  2. Raises awareness about possible limitations of conclusions based on beyond-Luttinger-liquid analyses which neglect the momentum-dependence of the interaction.

Weaknesses

  1. Section 5 assumes a very detailed familiarity of previous work [Refs. 7 and 8] on the effects of the momentum dependence of interactions in Luttinger-liquid-type models. A somewhat more detailed exposition of the key findings of these previous works would greatly aid the readibility of Section 5.

Report

This paper by Markhof, Pletyukhov and Meden (MPM) addresses two issues in the literature on one-dimensional correlated Fermi systems 'beyond the Tomonaga-Luttinger liquid paradigm', i.e. including the effects of band curvature.

A.
The first issue involves the computation of the dynamic structure factor, $S(q,\omega)$, of interacting one-dimensional spinless fermions with a nonlinear dispersion relation, by Pustilnik, Khodas, Kamenev and Glazman (PKKG) in 2006 [Ref. 17]. These authors pointed out that when computed to first order in the two-particle interaction, a logarithmic threshold singularity is obtained. Interpreting this term as the linear one in an expansion conjectured to sum to a power law, PKKG were able to deduce the exponent of the power law. MPM have now computed the second-order term explicitly and showed that it is consistent with the conjectured power law. This demonstration of consistency is reassuring, though not surprising (had they found an inconsistency, that would have been shocking...). I see the main value of their calculation in its detailed exposition (presented in an appendix), which very instructive: the authors explain carefully how the leading divergence can be extracted cleanly by a suitable shift of integration variables.

B.
The second issue involves a broader question. PKKG introduced an effective 'mobile impurity model', constructed in such a manner that it reproduces the power law in the dynamic structure factor. In 2007, PKKG [Ref. 22] used this mobile impurity model to compute the single-particle spectral function, $\rho(k,\omega)$, obtained via double Fourier transform of the single-particle Green's function $G(x,t)$. They found that $\rho(k,\omega)$ displays a power-law singularity on the hole mass shell, similar to that in the Luttinger liquid, and argued that this power law is universal. By contrast, in Section 5 of the present paper, MPM cautiously raise doubts about the latter statement. Let me attempt to summarize their argumentation, which builds on earlier papers by Meden and coworkers [Ref. 7 and 8], as follows:

(a) The standard Tomonaga-Luttinger liquid description of interacting 1D fermions is obtained by (i) linearizing the dispersion relation and (ii) neglecting any momentum dependence in the two-particle interaction vertex (i.e. assuming a point-like interaction). The resulting model famously is exactly solvable using bosonization. When used to compute the single-particle Green's function, $G(x,t)$, and from that the momentum-integrated single-particle spectral function, $\rho(\omega)$, a power law divergence is obtained for the latter, with an exponent which is 'universal', in that it depends only on the Luttinger parameters $K_\rho$ and $K_\sigma$.

(b) In 1999, Meden [Ref. 7] argued that this universality is lost when assumption (ii) is relaxed and the two-particle interaction is allowed to depend on momentum, $V(q)$ (as needed, e.g., to describe a finite-ranged interaction): Meden showed that then the power-law divergence of the single-particle Green's function, $G(x,t)$, involves exponents which do not depend only on $K_\rho$ and $K_\sigma$ [see Eq. (15) of Ref. 7], and argued that this implies the same for $\rho(\omega)$.

(c) In 2016, Markhof and Meden [Ref. 8] revisited this issue by considering the momentum-resolved spectral function, $\rho(k,\omega)$, for a Tomonaga-Luttinger-liquid model with linear dispersion but momentum-dependent interaction, $V(q)$. They obtained analytic expressions for $G(x,t)$ as a power series expansion in $z = e^{i (2 \pi/L)x}$, with $t$-dependent coefficients defined via recursion relations [see Eq. (43) of Ref. 8], and used these expressions to compute $\rho(k,\omega)$ numerically [see Fig. 2 of Ref. 8]. A careful analysis of the near-threshold behavior when $|k-k_F|$ is finite showed that for some choices of $V(q)$, no clear power-law behavior is found [see purple lines in Fig. 2 of Ref. 8, corresponding to V(q) decaying exponentially with $|q|$, cf. Eq. (55)]. Markhof and Meden summarize this finding by stating: "We provide strong evidence that any curvature of the two-particle interaction at small transferred momentum destroys power-law scaling of the momentum-resolved spectral function as a function of energy."

(d) Finally, in section 5 of the present paper, MPM argue that they expect the findings from (c) to apply also for a Tomonoga-Luttinger model with NONlinear dispersion and momentum-dependent interaction: the latter might destroy power-law scaling of the momentum-resolved spectral function as a function of energy. MPM do not attempt to substantiate this statement with explicit calculations, leaving this as an open issue for further investigations: "The e?ffect of band curvature on correlation functions of 1d interacting Fermi systems beyond the low-energy scaling limit (in which the curvature is RG irrelevant) remains an open issue which deserves further investigations."

In my view, the fate of power-law behavior in beyond-Luttinger-paradigm model - including both curvature of the dispersion and $q$-dependence of the interaction - is indeed worthy of further investigations. In that sense, I find the discussion of section 5 of the present paper reasonable. However, I would urge the authors to spell out the findings of Ref. 7 and Ref. 8 in a bit more detail in the early parts of their Section 5, to aid readers not thoroughly familiar with those papers in following their line of argumentation.

In particular, MPM should comment on the following possible objection to Refs. 7 and 8: The starting assumptions of the model of Refs. 7 and 8, namely (i) a linearized dispersion and (ii) a momentum-dependent interaction, seem to be mutually inconsistent, since the latter will, already at Hartree-Fock level, cause the effective dispersion to become nonlinear. Thus, once one has decided to linearize the dispersion, one should also neglect the momentum dependence of the interaction. According to this perspective, Refs. 7 and 8 considered a 'bad model', hence their conclusions may be disregarded.

My take on this point is: First, I have no reason to doubt the technical aspects of the computations from Refs. 7 and 8. Second, the possible objection just raised against the model considered in Refs. 7 and 8 does not apply for (d) below, which involves a combination of non-linear dispersion and momentum-dependent interaction. Hence, MPM's suggestion that the lessons from Refs. 7 and 8 might also apply to (d) strikes me as reasonable and worthy of further investigation.

I thus recommend the paper for publication, provided some comment on the issues raised in the preceding two paragraphs are included.

Requested changes

See last line of the above report.

  • validity: top
  • significance: ok
  • originality: ok
  • clarity: high
  • formatting: perfect
  • grammar: excellent

Author:  Lisa Markhof  on 2019-06-07  [id 534]

(in reply to Report 2 on 2019-05-31)

The referee nicely summarizes our results and intentions when mentioning the strengths of our paper.

The referee is right that we do not give a detailed introduction to Refs. [7] and [8] in Sect. 5. However, the main results of these papers of relevance for the present submission are summarized in the Introduction, Sect. 1. We believe that this is useful as it allows us to put our second main result (inconsistency of crucial parts of the momentum dependence of the interaction and the mobile impurity model) in a proper perspective already in the Introduction. In the light of the third report we are facing a conflict. While the second referee asks us to extend on the summary of earlier work the third one criticize our summary as too detailed. We thus hope that the second referee can go along with our decision to only slightly modify the discussion of Refs. [7] and [8] (as explained in the next two paragraphs).

We would like to emphasize that we do not agree with the referee that the model of Refs. [7] and [8] is a "bad model" with respect to the conclusions drawn from the calculations of these references. The goal of these papers was to present a counterexample of a model which is believed to be part of the Tomonaga-Luttinger liquid universality class, but does not show any universal power law at $k-k_F \neq 0$. We trust that the referee agrees that the Tomonaga-Luttinger model (linear single-particle dispersion) even with momentum dependent interaction should be part of this universality class. From this we concluded that the power laws of the single-particle spectral function found at $k-k_F \neq 0$ for the box potential or within the commonly employed ad hoc regularization are not part of the Tomonaga-Luttinger liquid universality. This is fully consistent with renormalization group arguments pointing at universality that can only be employed if all scales, including $k-k_F$, are sent to zero. We have added a corresponding comment in our paper.

We do not understand why the referee denotes the linear single-particle dispersion (of the fermions) and the momentum dependence of the two-particle interaction as being mutually inconsistent. We agree that in first order perturbation theory for the self-energy the fermionic dispersion could become nonlinear if the a momentum dependence of the two-particle interaction would be kept. The same holds for the bosonic dispersion within the bosonization approach in which the interaction is kept to all orders. We, however, fail to see how this indicates any inconsistency. We added a comment on this to our paper. Note, however, that this issue is irrelevant in the context of the present paper in which the combination of momentum dependence and nonlinearity of the single-particle dispersion is considered. This was already pointed out by the referee.

Anonymous on 2019-06-22  [id 544]

(in reply to Lisa Markhof on 2019-06-07 [id 534])

I am satisfied with the authors' replies to my comments and believe the paper is ready for publication.

Side remark: the second report of referee 2 also addresses an issue I had raised in my first report: namely that a $q$-dependence of the interaction generates, at mean-field level, a curvature for the dispersion, so that a model assuming linear dispersion but $q$-dependent interaction might be argued to be internally inconsistent. Given the contentious nature of this matter, the authors would be well advised to address this matter head-on in their paper -- formulate the potential objection, ideally using the phrasing of referee 1, and explain their reasons for rejecting it. (I write this as advice, not as a referee requirement.)

---

## Round 1 · Referee Report · Anonymous (Referee 3) · 2019-6-1

Strengths

The second order expansion of the dynamical structure factor is non-trivial and original

Weaknesses

The original results of the paper are not enough to warrant publication or substantiate the conclusions/claim of the paper.

Most of the paper is written more as a review than as paper focused on its own results.

Report

This paper computes the dynamical structure factor (DSF) of interacting spinless fermions, in a perturbation expansion of the interaction. Its goal is to analyze the behavior close to the threshold, and to compare with the predictions of the so-called non linear Luttinger liquid (NLL), of a modified, interaction dependent exponent close to threshold.

I have no problems with the derivation given in the present paper, leading to the new result of equation (22) but I don't think that the paper is suitable, at least in its present form to be published in Scipost, for the following reasons:

  • The goal of the present paper is to ascertain whether the claim of universality for the behavior close to threshold (essentially introduced in the references [6,17] of the present paper) are valid or not. In that respect the paper fails to give an answer on that point. The conclusion of the authors is that the interaction expansion is compatible with the expansion of a powerlaw (and thus that would be compatible with the claims of NLL) but that taking into account the momentum dependence of the interactions might lead to a different behavior. The might'' comes from the discussion around (27). To quote the authorsTo summarize, we have neither been able to find a way to keep the full momentum dependence of V nor found a justification for partially neglecting it''. So considering the results of the present paper (namely the second order expansion of the DSF) I feel that there is not enough material to justify publication in the context of the goal of the present paper (namely the test of the NLL). If the authors want to publish their results purely as a second order calculation (which is indeed non trivial) then this is of course possible but the paper must be rewritten in quite different way.

  • The bulk of the paper is more written as a review paper, or as a pedagogical presentation or a discussion on the question of the exponents of the DSF, than a paper presenting original research results. Most of the material before section 3 is a summary of well known facts on the TLL. Section 4 is mostly a reminder of the impurity model. Section 5 does not contain new results or does not discuss the results of the present paper but discusses the various papers in the literature (exactly solvable models, numerical calculations, etc.) and whether these works are conclusive or not for the predictions of the NLL. The conclusion of the paper is ``We conclude that more research is required to verify the predictions obtained with the mobile impurity model, in particular, for correlation functions other than the dynamical structure factor''. So although this could be a nice paper in the context of lecture notes of a school, I would not recommend publication as an original research article.

In conclusion, I feel that the present paper does not substantiate with enough original material the goals that are announced in the title/introduction. The derivation of the second order expansion of the DSF which is non-trivial and original could be published independently (equivalent of a brief report) in another context.

  • validity: low
  • significance: low
  • originality: low
  • clarity: high
  • formatting: perfect
  • grammar: excellent

Author:  Lisa Markhof  on 2019-06-07  [id 535]

(in reply to Report 3 on 2019-06-01)
Category:
reply to objection

There seems to be a misunderstanding about the goal of our work. We do not claim that we can make any definite statements about the earlier conjectured universal power-law behavior close to the threshold, even when just focusing on the dynamical structure factor (DSF). This holds even more so when it comes to other correlation functions, such as, e.g., the single-particle spectral function. In the title, abstract, and introduction we only promise to investigate the roots of the nonlinear Luttinger liquid phenomenology in several ways. We do exactly this. Some of our results support the conjectured behavior, others question the model--the mobile impurity model--which was argued to be the effective model of the nonlinear Luttinger liquid phenomenology. Our work is thus intended to stimulate a discussion on the roots and the results of the nonlinear Luttinger liquid phenomenology which we believe to be overdue. This holds in particular, as, even after years of research, convincing results confirming the conjectured behavior from other considerations, besides computations within the mobile impurity model, are missing. For details on this, see our discussion of Sect. 5.

We do not regard our analysis of the approximations leading to the mobile impurity model presented in Sect. 4 as review material. Instead, it is a critical assessment of this "derivation" which cannot be found anywhere in the literature. We believe that a major part of the community is unaware of the heuristic nature of many of the crucial steps as this does not become clear from the original literature. We emphasize that several of these steps are not backed-up by general arguments, of, e.g., RG nature or similar. This has to be contrasted with the "construction" of the Tomonaga-Luttinger model as the effective low-energy model of the Tomonaga-Luttinger liquid universality class (even neglecting the momentum dependence for that matter). We emphasize that none of the referees doubted that our assessment of the "derivation" is correct. We have modified Sect. 4 to make this more clear.

We have furthermore rewritten the Abstract and revised the Introduction to make more obvious that we present two novel results: (1) The consistency of second order perturbation theory with the conjectured threshold power law of the dynamical structure factor (DSF) but (2) the incompatibility of a crucial part of the momentum dependence of the two-particle interaction and the mobile impurity model.

The referee's summary of our work also indicates a misunderstanding about our second order calculation for the DSF. In this all the momentum dependence of the two-particle interaction, which is relevant for the leading behavior close to the threshold, is fully considered. Doing so the second order contribution is consistent with the conjectured power law and in this part of our considerations there is no "might" concerning the momentum dependence (only concerning terms to third order and higher). We only question whether the treatment of the momentum dependence of the two-particle interaction is justified when it comes to the construction of the mobile impurity model which was used in the literature to compute other correlation functions. The changes in the Abstract, the Introduction and Sect. 4 are also intended to avoid this misunderstanding.

Also Sect. 5 should not be viewed as a review. Instead, we show that no fully convincing results confirming the conjectured universality have been obtained by other means than computations within the mobile impurity Hamiltonian. Again, such a clear evaluation cannot be found in the literature.

To summarize, we present two original results. The second order results for the DSF, characterized as being nontrivial and important by all three referees, as well as the insight that it is impossible to keep crucial parts of the momentum dependence of the two-particle interaction when constructing the mobile impurity model. We believe that this is sufficient to justify publication. On top, our paper is intended to initiate an overdue discussion on the foundations of the nonlinear Luttinger liquid phenomenology. We believe that SciPost is an appropriate platform to do so.

---

## Round 2 · Referee Report · Anonymous (Referee 1) · 2019-6-15

Strengths

Explicit calculation of the higher-order correction to an observable in the Appendix to the main text.

Weaknesses

Misleading conclusions based on the erroneous previously-published works of the same authors and vague inconclusive discussion presented in the current manuscript.

Report

The response to the previous Report solidifies my opinion that the Authors harbor misconceptions about basics of one-dimensional quantum fluids theory. The conventional Luttinger liquid theory does provide the correct phenomenology of a generic 1D quantum fluid (with a nonlinear spectrum of constituent particles) for certain observables and in a certain limit. The validity of Luttinger liquid theory is illustrated by rather than based on an exactly solvable model.

The nonlinear Luttinger liquid specifies the limit in which the linear theory works. It makes rather minimal extension of the theory to account for the effect of curvature of the single-particle spectrum on the observables close to the threshold singularities (which presence is protected by kinematics of particles in 1D). The preservation of the power-law singularities is warranted by the same general principles which warrant the existence of commonly-known Fermi edge singularities beyond the theory of free fermions.

A correct detailed microscopic theory for the spectral function would be of value. However the Authors' papers preceding the current submission are flawed in a pretty basic part, as the linear dispersion is unstable with respect to the momentum-dependent interactions. One has to account, e.g., for the Harteree-Fock corrections which would curve the dispersion relation and break the spurious degeneracy of the many-body spectrum inherent for the linear dispersion. The Authors seem to be oblivious to that, coming to entirely wrong conclusions in their previous work. The main text (not the Appendix) of the submitted one is no better. In my view, its publication would do further damage to the field.
  • validity: poor
  • significance: poor
  • originality: -
  • clarity: -
  • formatting: -
  • grammar: -

Author:  Lisa Markhof  on 2019-06-15  [id 541]

(in reply to Report 1 on 2019-06-15)
Category:
reply to objection

It is rather difficult for us to reply to this report. The referee mainly, actually almost exclusively, criticizes (partly in the first paragraph but mainly the third one of the report) an earlier paper of two of us (LM and VM). Needless to say that we do not agree with the very vague arguments against this earlier work. In any case this earlier work is not the topic of the present reviewing process.

However, the first paragraph shows that it is not us, but the referee who suffers from misconceptions of Tomonaga-Luttinger theory. The crucial argument to show universality is the RG irrelevance of the band curvature and the momentum dependence of the two-particle interactions. In an RG procedure the Tomonaga-Luttinger model is the fixed point. It stands at the heart of Tomonaga-Luttinger liquid universality and not only illustrates this concept. In contrast to the referee, in our manuscript we clearly spell out the observables which are part of the Tomonaga-Luttinger liquid universality. Most crucially it only applies if RG arguments can be used, that is if all energy scales vanish.

In the second paragraph the referee just repeats the standard arguments used to support the mapping to the mobile impurity model. The referee does not even try to argue against our detailed analysis of the weaknesses of the mapping.

We believe that the last sentence of the report shows that the referee is highly biased and not open to scientific arguments which indicate weaknesses in the basis of the nonlinear Luttinger liquid phenomenology. Just take the fact that the referee ignores that any search for this phenomenology in microscopic models was so far not successful (for details see our manuscript).

---

## Round 2 · Referee Report · Anonymous (Referee 3) · 2019-7-5

Strengths

Second order perturbative calculation of the structure factor

Weaknesses

A (too) large part of the paper has a review character and does not really present original results

Report

In this revised version the authors have improved some aspects of the presentation of the results. The results themselves and the message of the paper are essentially unchanged.

As mentioned in the previous report the main, and in my opinion, only new result of the paper is the calculation of the perturbative expansion (up to second order) of the structure factor. This calculation is sound and constitute a new result that would be worthy of publication by itself, at the level of a "brief report" (the editor of Scipost should fix the threshold level they want for the papers published in Scipost).

However, the authors want to put this result in the more general context of discussing the validity and assumptions behind the non-linear Luttinger liquid model/derivation that exists in the literature. This is done in several stages:

  • the introduction (Section 1) contains a reminder (recalling the results of papers [7,8]) that for systems with momentum dependent interactions one does not find the TLL powerlaw behavior of one is at a finite energy scale (e.g. by having kkF0. It then mentions the impurity model that was used to address this finite energy scale question.

  • the Sections 2 and 3 contain the explicit perturbative calculation of the structure factor. The result is found consistent with the expansion of the powerlaw behavior predicted by the impurity model (up to second order of course).

  • Section 4 is reminder/rederivation of the impurity model (essentially introduced in e.g. Ref [6]). This section reproduces essentially the steps of Ref.[6] pointing out explicitly (for example just below equ. (26)) where approximations were made in the initial derivation. A summary of these is given at the end of the section (points (i)-(iv)). In my opinion this part does not contain new results but is simply a guided derivation of the model of Ref[6].

  • Section 5 contains a discussion of the various results that exist in the literature concerning the impurity model with the conclusion that none of them is conclusive to justify the universal result claimed by the model. Again in this section there is no original contribution by the authors, except a critical reading of the literature.

The final conclusion of these two sections is largely inconclusive, with the fact that "more research is needed" to decide if the impurity model has indeed a universal behavior or not.

I am not contesting the warnings that are made in these two last sections, but I reiterate my opinion that Sections 4 and 5 are not presenting original results but more a critical reading and derivation of existing material in the literature. I would be perfectly happy to see such chapters in e.g. notes of a school, but I am uneasy to see this as a sizable part of an original research article, moreover in a way that is essentially decoupled from the part that contains the original result of the paper.

I realize that the latter is subjective and different people might have different thresholds for what is considered as the content of an original research paper. For my part I think that the paper would benefit from seriously compressing Sections 4 (for example one could keep mostly the points (i)-(iv)) and Section 5, to provide a single section of discussion that would follow the results of the second order expansion (consistent with the predictions of the impurity model) with some comments of caution on the model.

Requested changes

  • strong compression of the Sections 4 and 5

---

## Round 3 · Referee Report · Anonymous · 2019-8-20

Strengths

1-The second-order calculation of the dynamical structure factor is technically challenging and the result nicely confirms the expectation of a threshold singularity.

Weaknesses

1-The criticism of nonlinear Luttinger liquid phenomenology is vague and based on a misunderstanding of the validity of the approach.

Report

The point of this manuscript is to put the nonlinear Luttinger liquid theory to the test. First, the authors tested the prediction of a threshold singularity in the dynamical structure factor (DSF) calculated within an effective mobile impurity model by Pustilnik et al. in Ref. [17]. They find that the perturbative calculation of the DSF to second order in the fermionic is consistent with the threshold singularity. This is the only result of the manuscript.

The authors claim to have a second result which is actually a remark about their own difficulty in handling the momentum dependence of the interaction within the mobile impurity model. This is not a serious objection to the nonlinear Luttinger liquid theory because it is based on a misunderstanding of the regime of validity of the mobile impurity model. As clearly stated in the original paper by Pustilnik et al., the definition of the impurity subband centered at momentum kF-q requires a momentum cutoff which is much smaller than q. This is important to distinguish it from the low-energy subband of right movers centered at momentum kF. Such procedure is devised to compute the threshold behaviour for correlation at small but finite q. As q decreases, the energy window in the DSF where the effective model can be applied shrinks, and one observes a crossover to the power law behavior of the conventional Luttinger liquid theory. In this manuscript the authors state that "the scale on which the above procedure, if applicable at all, is valid remains open", but the energy scales involved and the resulting crossover have been discussed for the spectral function in Imambekov and Glazman, Science 323, 228 (2009); see also the review in Ref. [6].

As a consequence of the momentum cutoff in the impurity subbands, it only makes sense to use the effective model in the regime where k1, k2, k3 in Eq. (25) are much smaller than q. One cannot take k1 \approx q as the authors do. If one really wants to go beyond the leading approximation, the correct procedure would be to expand the interaction potential in the small momenta within the subbands, generating additional interactions with higher powers of momentum. The authors seem to be bothered by the fact that an impurity model which is not restricted to density-density interactions is no longer solvable. However, such additional terms are allowed by symmetry even in the original context of the x-ray edge singularity in metals where the original microscopic model is not Galilean invariant. The general expectation is that they may renormalize the parameters of the effective theory, but do not remove the power law behavior in the cases where a finite-frequency lower threshold at finite q is guaranteed by kinematic constrains (see Khodas et al., Ref. [27]). If the authors really want to point out a problem with the mobile impurity model of the nonlinear Luttinger liquid theory, they should show that the perturbations which are higher order in momentum destroy the threshold singularity. However, this seems rather unlikely.

In their concluding remarks, the authors also manifest their skepticism of previous papers that provided evidence for the nonlinear Luttinger liquid phenomenology. Of course numerical calculations such as those in Refs. [24] and [25] have limited frequency resolution, but together with the exact solutions of integrable model they do support the whole picture. Even if the authors want to leave out long-range potentials as in the Calogero-Sutherland model, they should acknowledge the example of the Lieb-Liniger model in Kitanine et al., J. Stat. Mech., P09001 (2012), in which the exact calculation of form factors are consistent with the nonlinear Luttinger liquid theory. Judging by what has been presented in this manuscript as compared to the existing literature on this subject, I don't find it reasonable to conclude that "this raises doubts that the conjectured nonlinear Luttinger liquid phenomenology can be considered as universal", as stated in the abstract.

Requested changes

1- Below Eq. (24), the authors should mention the interpretation for the q\to0 limit within the nonlinear Luttinger liquid theory; cite Imambekov and Glazman, Science 323, 228 (2009).
2-Please remove statements about the momentum dependence with k\approx q around Eq. (25).
3-The criticism of Refs. [24-29] and the conclusion are too biased and unjustified. Even if the authors believe that more research is required, they should at least acknowledge the successful results of nonlinear Luttinger liquid theory. I recommend rewriting the conclusion and the last statement in the abstract.

  • validity: ok
  • significance: ok
  • originality: ok
  • clarity: high
  • formatting: perfect
  • grammar: perfect

Author:  Volker Meden  on 2019-08-27  [id 585]

(in reply to Report 1 on 2019-08-20)

Although the new referee is also critical about our second new result,
in contrast to the first referee, the present one provides scientific
arguments. We are very glad about this as it gives us the opportunity to
argue in an equally scientific way that the criticism is based on a
misunderstanding of our second result and a notation issue.

The main problem is that the referee overlooks that our second new result
is based on two ingredients. The first is that we have shown earlier in
Refs. [7,8] that for a strictly linear single-particle dispersion
(of the fermions) any nonconstant momentum dependence of the
two-particle interaction close to momentum transfer q=0 destroys
power-law (threshold) scaling of the single-particle spectral function
as a function of energy at any finite k-k_F. Dropping the momentum
dependence leads to spurious power laws. In the present manuscript we
show that exactly this type of momentum dependence cannot be kept
when attempting to map the interacting electron gas onto a solvable
mobile impurity model. One can thus not exclude that in analogy to the
Tomonaga-Luttinger model this leads to spurious power laws. In the report
the referee ignores the first part of the argument leading to our final
conclusion that more work on the combined effect of a nonlinear fermionic
single-particle dispersion and a momentum dependent two-particle
interaction is required. In fact, in the reports of three of the four
referees of the different versions of our submission we find indications
that they did not seriously consider this part of our argument (the
exception being the second referee of version 1 of our submission).

We next go into more details. In the second paragraph of the report
the referee states that "As q decreases, the energy window in the DSF
where the effective model can be applied shrinks, and one observes a
crossover to the power law behavior of the conventional Luttinger liquid
theory." We have two problems with this sentence. Firstly, in the
Tomonaga-Luttinger model, being the fixed point model of the
Tomonaga-Luttinger liquid universality class, the DSF at small momenta
q does not show any power-law behavior but is rather given by a
delta-function. The referee might have confused the DSF and the
single-particle spectral function. In case the referee does indeed speak
about this spectral function we, secondly, have the problem that there
is nothing like "...the conventional Luttinger liquid theory..." result
for this, at least as long as the (fixed) momentum is different from k_F.
This is what we describe above and have shown in Refs. [7,8]. Therefore,
in the present case not even the limit of small (but finite) momenta is
'universal', or more precisely, the single-particle spectral function
depends on details of the momentum dependence of the two-particle
interaction (which are not kept in the the nonlinear Luttinger liquid
phenomenology). Do the protagonists of the nonlinear Luttinger liquid
theory seriously believe that the single-particle spectral function
shows for sufficiently large fixed |k-k_F| a 'universal' power-law
behavior (as a function of energy) with a momentum dependent exponent,
becomes nonuniversal (no power law; see Refs. [7,8]) for smaller
|k-k_F|, and, eventually, universal again at k-k_F=0 (in the sense of
conventional Tomonaga-Luttinger liquid theory; see Refs. [7,8])? If so
they should show this. This was clearly not done so far, mainly because
the absence of universality of the single-particle spectral function
of the Tomonaga-Luttinger model, or, more generally, of
Tomonaga-Luttinger liquids, at finite k-k_F was mainly ignored. We
furthermore wonder what the origin or nature of the 'universality' at
finite k-k_F might be? In contrast to the Tomonaga-Luttinger liquid
universality, emerging when all energy scales are sent to zero, it is
clearly not based on powerful RG arguments.

In the third paragraph, the referee notes that k1, k2, k3 in Eq. (25)
are all much smaller than q, and criticizes that we would take
k1 \approx q. This is a misunderstanding due to a confusing notation
on our part. We apologize for this, and thank the referee for pointing
this out. k1 refers, at this point, to k1 in Eq. (11) rather than
Eq. (25). We have amended our notation to make clear which momentum we
refer to, and have also added comments for clarification. In this
context, we also noted that there was a typo in the paragraph above
Eq. (25): instead of "... can only be close to the momenta 0, \pm kF,
..." it should be "... 0, \pm q ...". We have corrected this. Just as
in the original considerations of Pustilnik et al., we take the
momentum cutoff of the subbands to be much smaller than q. However, no
calculations can be found in the literature which show that dropping
the momentum dependence, as it is done when 'deriving' the mobile impurity
model, is a controlled approximation, even when focusing on this rather
special limit. In particular, no (powerful) RG arguments, often used when
the 'irrelevance' of perturbations is to be investigated, are available.
Our experience from Refs. [7,8] taught us to be extremely careful when
it comes to dropping the momentum dependence without having rigorous
arguments to do so. To be more specific, even if k_1' in Eq. (25) (revised
notation) is taken much smaller than q (see the third paragraph of the
report) it is by no means obvious that replacing V(q-k_1'+k_2'-k_3') -> V(q)
is a controlled step. All this is strongly linked to the question in which
sense the terms kept in the 'derivation' of the mobile impurity model
are 'the leading ones'. In fact, in the third paragraph the referee
also uses this phrase ("If one really wants to go beyond the leading
approximation..."). We believe that 'the leading ones' should be replaced
by 'the ones which can be kept when aiming at a solvable Hamiltonian'.
As we have mentioned in earlier replies our experience shows that this
weakness of the 'derivation' of the (solvable) mobile impurity model is
not recognized by a sizable part of the community.

Another argument put forward in the third paragraph of the report is
"The general expectation is that they may renormalize the parameters of the
effective theory, but do not remove the power law behavior in the cases
where a finite-frequency lower threshold at finite q is guaranteed by
kinematic constrains (see Khodas et al., Ref. [27]). ... However, this
seems rather unlikely." We emphasize that the referee here has to use
phrases such as "general expectation" and "seems rather unlikely". The same
phrases were used when the effect of the momentum dependence of the
two-particle interaction on the single-particle spectral function of the
Tomonaga-Luttinger model (strictly linear fermionic single-particle
dispersion) was discussed in the past. However, Refs. [7,8] showed that
the general expectation is wrong and the unlikely case is realized in
this model. In analogy to the present case it was obvious that (powerful)
RG arguments cannot be used to argue in favor of universality (as an
energy scale, namely k-k_F, is kept fixed). We agree with the the referee
that a detailed analysis of the relevance of terms dropped when 'deriving'
the mobile impurity model from the interacting electron gas is missing.
We, however, disagree with the referee that we have to "...show that
the perturbations which are higher order in momentum destroy the
threshold singularity." In contrast, the protagonists of the nonlinear
Luttinger phenomenology should have shown already years ago that the
threshold power laws are stable against such terms. In our present
manuscript we take one step in this direction by showing that no
(exactly) solvable model in which relevant parts of the (bulk) momentum
dependence are kept can be derived in a straightforward way and analyze
the reason for this.

We emphasize, that we do not argue against any of the "classic" results
obtained for the x-ray edge problem; see the third paragraph of the
report.

In response to the requested changes we have revised the manuscript
as follows:

(1) We agree with the current referee that a comment on the papers by
Kitanine, Kozlowski, et al.. is important. In fact, an earlier version
of our submission contained such a comment which was, however,
criticized by the Editor. We made the mistake to remove it instead of
rewriting it. We now added a brief discussion on these papers (not only
J. Stat. Mech., P09001 but also arXiv:1811.06076) in the discussion
of Sect. 5 and hope that both the current editor and the current referee
are satisfied with this.

(2) In earlier rounds of reviewing we were asked by the referees and
the editors to shorten our discussion of Refs. [24-29] (Refs. [25-30]
of the revised version) with the argument that the weaknesses are well
known. We agree with the current referee that it might be advantageous
to extend this discussion which would allow us to be more specific.
However, we feel that given the history of our submission we cannot
return to the former version. In any case, we do not believe that
"The criticism of Refs. [24-29] and the conclusion are too biased and
unjustified." We simply mention these important papers and briefly state
their limitations.

(3) We have rewritten our comment on the q-> 0 limit following Eq. (24)
along the lines suggested by the referee.

(4) We have amended our notation in Eq. (25) and added comments to clarify
which momentum we refer to.

We hope that our manuscript can now be accepted for publication.

---

## Round 3 · Author Response

Dear Editor,

Thank you very much for reconsidering our submission and for your
feedback.

We have uploaded a revised version of our manuscript to arXiv.
We followed your and the former editor's advise to significantly
shorten sections 4 and 5. In fact, section 4 now only has less
than half the length it used to have and more or less exclusively
describes our observation that crucial parts of the momentum dependence
of the two-particle interaction cannot be kept in the "derivation" of
the mobile impurity model from the 1d interacting electron gas. This is
our second new result (the first one being the second order perturbation
theory for the DSF described in section 3 and the appendix). We only kept
a small part of section 5 and merged it with the old section 6 to a new
section 5.

Sections 1 to 3 of the revised submission only contain minor changes.
They were required for consistency reasons (with respect to the changes
in the other sections).

Concerning your more specific remarks (1) to (4).

(1) We do not completely agree with your statement that everybody in the
field is fully aware of the phenomenological nature and the weaknesses
of the nonlinear Luttinger liquid approach. In fact, when discussing
our new results with colleagues we frequently encountered exactly the
opposite. Many colleagues believe that with the advent of the nonlinear
Luttinger liquid phenomenology the issue of the nonlinearity of the
single-particle dispersion is settled. However, in our revised version
we express that certain parts of the community are aware that many steps
in the "derivation" of the mobile impurity model from the interacting
electron gas are at most phenomenological. In this respect we also

(2) included the reference you mentioned and frequently refer to it.

(3) We did not intend to indicate that it is not settled why the
Calogero-Sutherland model does not fall into the (alleged) nonlinear
Luttinger liquid universality class. With the revisions of section
5 this is no longer an issue.

(4) Also your remark concerning the old Ref. [25] (new Ref. [28]) is no
longer an issue.

As emphasized in an earlier conversation with the former editor of our
submission we do not doubt that the first referee has worked on 1d
correlated systems. However, this referee is highly biased and was
unwilling to engage in any scientific discussion. Instead, the second
report of this referee contains impudent statements. We would very much
appreciate if this would be acknowledged from the editorial side. On
general grounds we believe that reports of this type should simply be
dropped.

We hope that the revised version can now be accepted for publication
in SciPost.

Yours sincerely,

Lisa Markhof
Mikhail Pletyukhov
Volker Meden

---

## Round 4 · Referee Report · Anonymous (Referee 5) · 2019-9-10

Strengths

none

Weaknesses

Numerious. See report.

Report

The paper "Investigating the roots of the nonlinear Luttinger liquid phenomenology" by L. Markhof, M. Pletyukhov and V. Meden
discusses some aspects related to the universality of the nonlinear Luttinger liquid phenomenology.

The first part of the paper is devoted to a second order perturbation theory calculation for the dynamical structure factor in a model of spinless fermions
so as to test the validity of the nonlinear Luttinger liquid based predictions. Not astonishingly, the authors find a matching between their perturbative calculations and the
power-law behaviour near the lower-edge as predicted by the use of the nonlinear Luttinger liquid (NLLL).

The second part of the paper raises criticisms towards the nonlinear Luttinger liquid universality. The main back up for this criticism are results issuing from
old works of a subset of the authors, Ref. [7] and [8]. As such, the criticisms are thus very disconnected from the first part of the paper.
The authors try to argue that "single-particle spectral function"'s power-law behaviour might not be grasped by a nonlinear Luttinger liquid (or that it might not even exist!).
The strategy employed by the authors to defend their point is very strange. If they believe so, they should have simply computed the second order perturbation theory result for these spectral functions
and then check that it agrees or show explicitly that there is a discrepancy. Yet this is not done.

Instead, the author give some vague arguments and try to critically assess the existing literature on the subject. The author's main point is that the behaviour of the correlators in real space along the directions
$x \pm v t=0$, with $v$ the Fermi velocity, spoils the dynamic response functions' power-law behaviour on the edges of a given model's spectrum. However, it is a rather easy exercise to convince oneself that this behaviour may only affect
the behaviour of dynamic response functions close to edges of the spectrum's curve $(k,\mathcal{E}(k))$ whose dispersion relation $\mathcal{E}(k)$ satisfies to the constraint $\mathcal{E}^{\prime}(k)=v$.
These are very special points and I am not aware of anyone serious in the business ever claiming
that NLLL grasps the behaviour of dynamic response functions for these special cases. I will not elaborate further on the other criticisms of the authors of the mobile impurity model since
these issues were already discussed in the previous reports.

However, I would like to focus on the authors criticisms of the analysis carried out on integrable systems in the works [29,31].
This part is very mind blowing to me in that they criticise the content and results that are \textit{not} established in these works!
The authors write "the lattice model of spinless fermions with nearest-neighbor interaction, which is equivalent to the XXZ
Heisenberg model, was studied and the form factors of the model were examined analytically." However, as it is suggested from that paper's title "On singularities of dynamic response functions in the massless regime
of the XXZ spin-1/2 chain", the work actually deals with the XXZ chain. Moreover, no form factors are analysed there. In fact the starting point of that paper
is a series of multiple integral representation for the XXZ chain's response functions.

Then the authors write "After resummation, the correlation functions in the limit of large x and t were found to exhibit
power-law behavior." Again, this is not done at all in [29]. Rather, the work [29] develops a rigorous method allowing one to analyse directly
the behaviour in, the momentum-frequency plane, of the dynamic response functions starting from their series of multiple integral representations.
The next two-sentences "This in turn yielded power-law behavior in dynamical response functions,
and the exponents agreed with the nonlinear Luttinger liquid prediction. The calculation relies on two-dimensional asymptotic analysis in real space and time." are also nonsense.
The asymptotic analysis carried out in the work [29] does actually deal with multidimensional integrals.
The authors write "In particular, the special directions x = ±vt, with the appropriate renormalized velocity of the elementary excitations
v, are not considered separately." Again, the authors do seem to exhibit a certain lack of understanding of the paper's content. The two branches do have to be analysed jointly
(what is done in [29]) and it is their mutual interaction that does produce the power-law behaviour. The work [29] provides a precise control on the corrections and, in particular,
on the potential effects that could be induced from the real space correlator's behaviour along the two special lines $x = ±vt$. These are shown not to contribute to the leading
non-integer power-law behaviour close to the edges of the spectrum (be it single or multi species edges). Independently of a total lack of connection between the author's criticisms and
the content of [29], it sounds to me pretty strange to try to wave-off the results of an exact rigorous analysis by some heuristic like argument. Finally,
the sentence "The same type of two-dimensional asymptotic analysis is a crucial ingredient of the form factor approach to
the Lieb-Liniger model [31]." also does translate the author's missunderstanding of the work's content.

It is also important to stress that, thanks to the recent progress on the analysis of spectral functions in the XXZ chain, there is almost no room for the NLLL paradigm to fail.
Indeed, a consequence of the work K. K. Kozlowski et J. M. Maillet, {\it Microscopic approach to a class of 1D quantum critical models}, J. Phys. A: Math. $\&$ Theor. , {\bf 48}, 484004, (2015),
is that the matrix elements of local operators in a model falling into the universality class of a Luttinger liquid take a very specific form between low-energy states.
In their turn, as follows from decades of calculations carried out in quantum field theories or condensed matter physics, and by means of various approaches,
the matrix elements of such local operators taken between finite energy states are described by a density of form factors, in the large volume limit.
Taken these two facts and assuming a natural parametrisation of the model's spectrum in terms of particle species, one may repeat the reasonings explained in
K. K. Kozlowski, {\it On the thermodynamic limit of form factor expansions of dynamical correlation functions in the massless regime of the XXZ spin 1/2 chain.},
J. Math. Phys. {\bf 59} (9), 091408 (2018), so as to provide an explicit functional form for the form factor expansion of two-point functions in such a model.
I stress that with the facts given as above, there is no need for the model to be integrable. Of course, some of the building blocks of such a series will remain unknown, so that
only the overall functional form of a massless form factor series will be available in such a model. This is explained in broader details in the mentioned works.
However, it is the functional form of the massless form factor expansion that does fix the edge behaviour of the response functions. Indeed, this is the only input that is needed so as to implement the asymptotic analysis
carried out in [29].

To summarise, I do not think that the scientific discussion in the second part of the paper makes the work fit for publication, be it in SciPost or any other journal.
Thus I strongly recommend to reject the paper.

---

## Editorial Decision

resubmitted